# Overview of Catalysts with MIRA21 Model in Heterogeneous Catalytic Hydrogenation of 2,4-Dinitrotoluene

**Alexandra Jakab-Nácsa** [1,2], **Viktória Hajdu** [2,3], **László Vanyorek** [2], **László Farkas** [1,2] and **Béla Viskolcz** [2,3,*]

1    BorsodChem Ltd., Bolyai tér 1, H-3700 Kazincbarcika, Hungary
2    Institute of Chemistry, Faculty of Materials and Chemical Engineering, University of Miskolc, H-3515 Miskolc, Hungary
3    Higher Education and Industrial Cooperation Centre, University of Miskolc, H-3515 Miskolc, Hungary
*    Correspondence: bela.viskolcz@uni-miskolc.hu

**Abstract:** Although 2,4-dinitrotoluene (DNT) hydrogenation to 2,4-toluenediamine (TDA) has become less significant in basic and applied research, its industrial importance in polyurethane production is indisputable. The aim of this work is to characterize, rank, and compare the catalysts of 2,4-dinitrotoluene catalytic hydrogenation to 2,4-toluenediamine by applying the Miskolc Ranking 21 (MIRA21) model. This ranking model enables the characterization and comparison of catalysts with a mathematical model that is based on 15 essential parameters, such as catalyst performance, reaction conditions, catalyst conditions, and sustainability parameters. This systematic overview provides a comprehensive picture of the reaction, technological process, and the previous and new research results. In total, 58 catalysts from 15 research articles were selected and studied with the MIRA21 model, which covers the entire scope of DNT hydrogenation catalysts. Eight catalysts achieved the highest ranking (D1), whereas the transition metal oxide-supported platinum or palladium catalysts led the MIRA21 catalyst ranking list.

**Keywords:** hydrogenation; catalyst ranking; catalyst comparison; 2,4-dinitrotoluene; 2,4-toluenediamine





## 1. Introduction

Polyurethanes, referred to as urethanes, PUs, or PUR, are characterized by the urethane linking –NH–C (= O)–O–, which is established by the reaction of the organic isocyanate (NCO) groups and hydroxyl (OH) groups [1]. Due to their versatility and excellent mechanical, chemical, physical, and biological properties, they have a wide range of applications and a variety of uses, such as in appliances, automotive, construction, furniture, clothing, and the wood industries. Although the impact of COVID-19 has been startling, the global polyurethane market size was USD 56.45 billion in 2020 and it is projected to grow [2]. The rising demand for foams in furniture and in the construction industry has been driving the toluene diisocyanate (TDI) market growth.

TDI is one of the main materials of polyurethane production. TDI is produced in three different steps: the nitration of toluene, the dinitrotoluene hydrogenation to toluenediamine (TDA), and in the phosgenation of diaminotoluene. The general industrial process of TDA formation is the catalytic hydrogenation of dinitrotoluene in the liquid phase in the presence of a catalyst. Six isomers of TDA can be generated, but the major intermediate of TDI production is 2,4-toluenediamine (2,4-TDA).

In addition to the production volume and the versatility of its application, the industrial importance of TDA production is also shown by its patent history. A search on the Google Patents website using the keywords 'dinitrotoluene', 'hydrogenation', and 'toluenediamine' yields 400 patents that have been published since 1953 [3]. While the first patents in the 1950s described some general reaction conditions and some catalyst components, the newest patents provide much more detailed descriptions of multicomponent catalysts, their composition, and their preparation [4–10]. Although the latest patents

describe the high performance catalysts comprising activated metal, one or more auxiliary metals, and a special support material such as oxide [11], the most commonly used catalyst in the industry is the nickel catalyst [12]. Despite the fewer number of published scientific research papers [13], a high conversion and selectivity were achieved with the catalysts of many different formulations of nickel, platinum, or palladium on carbon, oxide, or zeolite supports [14–19]. However, in addition to the catalytic performance, sustainability parameters also play an increasingly important role in the chemical industry, such as reversibility and stability [20–22]. There are many steps between the fundamental research on catalysts to their industrial application. Nevertheless, new scientific findings are essential for the development of applied technological innovations if the new knowledge is to be used effectively [23].

The Miskolc Ranking 21 (MIRA21) model is a new, multi-step, functional mathematical method to extract the knowledge from the heterogeneous catalyst data through the catalyst characterization, comparison, and ranking of a series of catalysts [24]. In our previous work we discussed the method and application possibilities through the reaction of nitrobenzene hydrogenation to aniline. The ranking model applies a fifteen-parameter descriptor system to facilitate the comparison of the experimental and scientific publication results of a determined reaction to support catalyst development. The parameters of the descriptor system can be divided into four groups: catalyst performance, reaction conditions, catalyst conditions, and sustainability parameters. The model qualifies and ranks the catalysts based on these parameters.

This overview summarized the advances in the selective hydrogenation of dinitrotoluene to form toluenediamine, based on the catalysts used to carry out this process in the last 50 years. As the focus point of this work, we characterized and ranked 58 catalysts from 15 articles according to the MIRA21 model to make the systematic comparison of them.

Figure 1 describes the technological process:

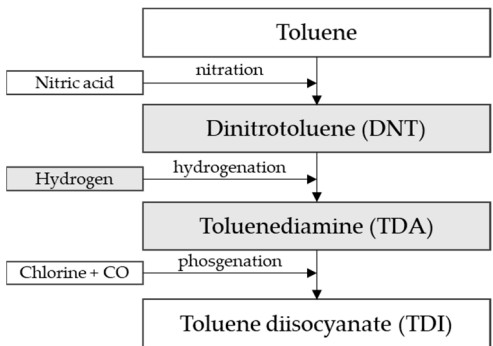

**Figure 1.** TDI production process from toluene [25].

The production of TDI is carried out in a three-step continuous process (Figure 1). Dinitrotoluene is produced in the first step by the nitration of toluene. The second and key step is the catalytic hydrogenation of dinitrotoluene to toluene diamine. In the last step, toluene diamine is phosgenated to form TDI.

The formation of DNT by the mixed acid nitration of toluene occurred at atmospheric pressure and between 40 °C and 70 °C. The main product of the process is a mixture of the 2,4- and 2,6-dinitrotoluene isomer mixture (Figure 2) [26]. These are the starting reagents for hydrogenation. The side products of the reaction are 2,3- and 3,4-DNT isomers, whereas the 2,5- and 3,5- isomers and other byproducts can also be found in smaller quantities.

**Figure 2.** Raw material of hydrogenation process [26].

The second step of the industrial process is the catalytic hydrogenation to dinitro-toluene to toluenediamine using a solid catalyst at a high pressure and high temperature (100–150 °C, 5–8 bar). This step was previously carried out in the presence of iron filings and aqueous hydrochloric acid [27], but today it is hydrogenated using a Ra-Ni or Pd/C catalyst. In strong industrial conditions (high pressure and temperature), an extremely high-quality product is produced with a high yield. Furthermore, Figure 3 shows the general reaction equation with the main product of dinitrotoluene hydrogenation.

**Figure 3.** General reaction equation with the main 2,4 isomer [27].

The process occurred in a continuously stirred tank reactor where the DNT isomer mixture usually reacts with hydrogen gas in the presence of the supported precious metal catalyst in a TDA/water medium. In order to achieve a high conversion, the correct catalyst composition and reaction conditions (temperature, pressure, etc.) are crucial [28]. The spent catalyst is removed from the system through a catalyst filter and the new catalyst is added. It is important that the catalyst can be easily removed and regenerated. Westerwerp et al. made a pilot installation of a 2,4-DNT synthesis plant and studied the reactor design and operation process [29–31]. The experiments took place in a continuously stirred, three-phase slurry reactor with an evaporating solvent. They mentioned that, in addition to a

good catalyst, it is important to choose the ideal hydrogenation reactor unit and optimal reaction parameters, and to solve the deactivation problem of the catalyst.

## 2. Reaction Mechanism and Kinetics

The kinetics and reaction mechanism of the catalytic hydrogenation of 2,4-dinitrotoluene to 2,4-toluenediamine was investigated by several research groups [15,16,32–34]. In the 1990s, Janssen et al. studied the reaction scheme and modelled the reaction rates and catalyst activity to evaluate the performance of a batch slurry reactor at 308–357 K and over the pressure range of 0–4 MPa [35,36]. The reaction rates are described by the Langmuir-Hinshelwood model. They found that 2,4-dinitrotoluene can be converted to 2,4-toluenediamine through two parallel pathways with consecutive reaction steps. They found that 4-hydroxyamino-2-nitrotoluene, 4-amino-2-nitrotoluene, and 2-amino-4-nitrotoluene are the most stable intermediates, but the presence of 2-amino-4-hydroxyaminotoluene and another azoxy compound were also observed. One of the reaction pathways is the direct conversion of an o-nitro group to an amino group. The other one is the conversion of the p-nitro group to an amino group in a two-step reaction.

While Janssen et al. was the first to describe the two reaction pathways, Neri et al. wrote a more complex reaction mechanism [37,38]. Neri et al. investigated this hydrogenation reaction over a supported Pd/C catalyst and found that 4-hydroxyamino-2-nitrotoluene, 2-amino-4-nitrotoluene, and 4-amino-2-nitrotoluene can form directly from 2,4-dinitrotoluene. Figure 4 compares the Janssen et al. and Neri et al. reaction schemes. The latter found that the hydrogenation of the hydroxylamine intermediate occurred via a triangular reaction pathway. Their further studies focused on 2-hydroxyamino-4-nitrotoluene as a reaction intermediate, which accumulates in the reaction mixture instead of 2-hydroxiamino-4-nitrotoluene [37,39,40]. It was shown that the formation of the nitro group depends on the presence of electron-donating substituents and steric effects [41].

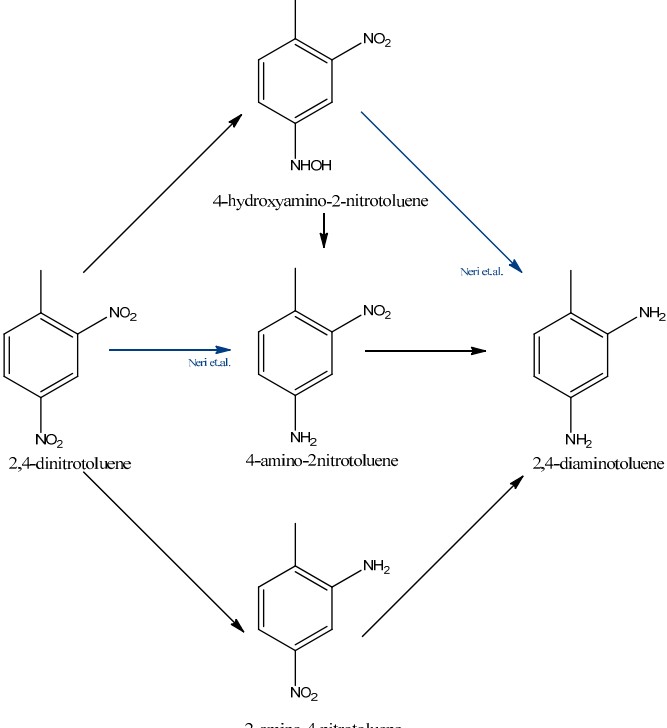

**Figure 4.** Dinitrotoluene hydrogenation pathways according to Janssen et al. and Neri et al. (Blue lines shows the new pathways comparable to Janssen et al.'s results.) [37].

Electronic structure computational studies can be a great help in studying the reaction mechanism of three-phase catalytic hydrogenation reactions. In such a study, Barone

et al. applied the Monte Carlo algorithm to simulate the batch hydrogenation of 2,4-dinitrotoluene on a carbon-supported palladium catalyst [31,42,43]. They investigated the influence of the molecular adsorption modes, the steric hindrance, and the metal dispersion on the reaction mechanism. They found that the steric hindrance of the different surface species had the largest influence on the mechanism.

Hajdu et al. worked on a new catalyst that contains precious metal on chromium-oxide nanowires for 2,4-toluenediamine synthesis [44]. In our previous work, we examined and described a possible reaction mechanism based on the GC-MS results. Our study confirms the mechanism by Neri et.al., as we detected the presence of nitroso and hydroxylamine compounds (Figure 5).

**Figure 5.** Reaction mechanism according to Hajdu et al. [44].

According to our results, 2,2-dinitro-4,4-azoxytoluene was found in the system (Figure 6), which could form through the reaction between the nitroso and hydroxylamine functional groups. In addition to the two semi-hydrogenated intermediates (4-amino-2-nitrotoluene, 2-amino-4-nitrotoluene), we detected other side products, which further supports the reaction mechanism of Neri et al. Figure 7 shows the detected and assumed side products obtained during the formation of TDA.

2,2-dinitro-4,4-azoxytoluene

**Figure 6.** Formation of 2,2-dinitro-4,4-azoxytoluene [44].

**A**

(E)-1-(2,4-dinitrophenyl)-N-(2-methyl-5-nitrophenyl)methanimine

**B**

2-[(E)-[(2-methyl-5-nitrophenyl)imino]methyl]-5-nitrophenol

**C**

4-[(E)-[(2-methyl-5-nitrophenyl)imino]methyl]benzene-1,3-diol

**D**

3-methoxy-4-[(E)-[(4-methyl-3-nitrophenyl)imino]methyl]phenol

**E**

N-(2,5-dimethylphenyl)-2-methoxy-4-nitrobenzamide

**F**

N-(3-methoxy-4-methylphenyl)-4-methyl-3-nitrobenzamide

**G**

(E)-1-(2-methoxy-4-methylphenyl)-N-(3-methoxy-4-methylphenyl)methanimine

**Figure 7.** Possible side products of the TDA synthesis according to Hajdu et al.'s research (black line—detected molecules, blue line—assumed molecules) [44].

We also demonstrated that E-1-(2,4-dinitrophenyl)-N-(2-methyl-5-nitrophenyl) methanimine and 2-[(E)-[(2-methyl-5-nitrophenyl)imino]methyl]-5-nitrophenol was formed by the water loss in the condensation reaction (**A** and **B**). Molecule **C** was formed by the reaction between 4-methylbenzene-1,3-diol and 2-nitroso-4-nitrotoluene. As shown in Figure 7, 2-nitro-4-nitrosotoluene reacted with 2-methoxy-4-methylphenol to yield compound **D**. Isomers **E** and **F** were formed by the reaction between 2-methoxy-1-methyl-4-nitrobenzene and dimethyl-2-nitrobenzene. Compound **G** was formed by the reaction between 2-methoxy-1,4-dimethylbenzene and 2-methoxy-4-nitrosotoluene.

## 3. Results and Discussion of TDA Synthesis Catalysts

The hydrogenation of 2,4-dinitrotoluene to 2,4-toluenediamine is an essential technological step in the polyurethane industry. Although the technological process, the reaction mechanism, and the reaction kinetics have been investigated and have come to be generally accepted, there is still much to learn about the catalysis of this process. That is why the mapping of the current state of catalyst development likewise facilitates the development

of scientific research. However, the review of the literature on catalysts used for TDA synthesis does not provide sufficient information to achieve this aim. The comparison of the catalysts examined so far provides a much more comprehensive picture of the latest developments on their effectiveness. Therefore, the MIRA21 model was used to execute the catalyst's characterization, comparison, and qualification [24].

### 3.1. Catalyst Library

The results of the literature research are surprising because there are relatively few published scientific results about the dinitrotoluene hydrogenation process. They were mostly prepared before the 2000s. Based on Google Scholar searches for the keywords *dinitrotoluene hydrogenation*, we obtained 2210 matches, however, if we added *toluenediamine*, there were only 212 hits. In total, 92 pieces of these included scientific results obtained after 2010. To demonstrate this, the keyword *kinetic* was added to the initial search, which then yielded 120 articles. Overall, only a few research groups have studied TDA synthesis and have prepared catalysts for this reaction. On one hand, a smaller database reduces the reliability of the MIRA21 results. On the other hand, a smaller dataset makes it easier to delineate the possible research pathways on the topic.

After the first selection, 56 articles remained. During the data analysis, we concluded that it is justified to change the publication year selection criteria (after 2000) and we also worked with previous articles. The left panel of Figure 8 shows the distribution of the scientific publications according to the publication date. The right panel of the figure presents the studied articles based on its Q-index in 2021 after the primary article selection (relevance, publication year, Q-index). The figure shows that the data used to analyze the catalysts mainly came from Q1 articles. A few publications whose publisher has since ceased to exist were also included in the analysis because they had previously provided space for the publication of high-quality scientific works.

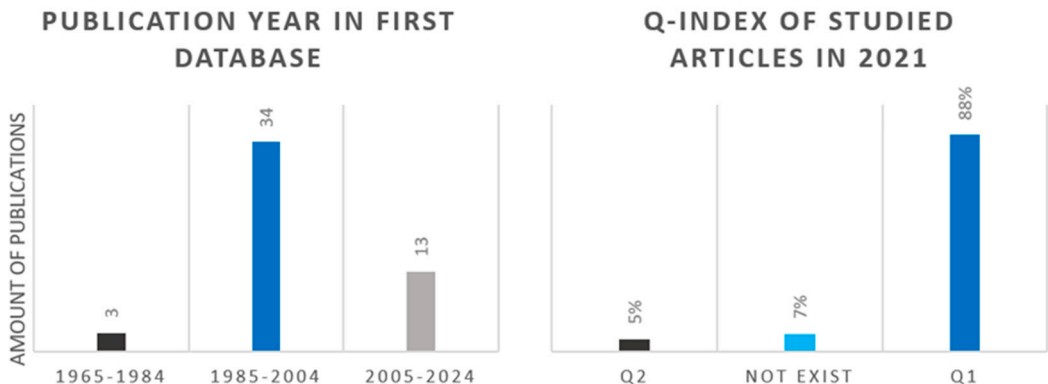

**Figure 8.** Publication year distribution of 56 articles after first selection (**left**) and Q-index distribution of 15 articles after second selection (**right**).

The 58 qualified catalysts selected from 15 articles were mostly supported catalysts (Figure 9 left) [14,16,40,44–55]. Most of the produced catalysts contained one active component on the support (middle of the figure). The catalysts with two active components generally applied palladium-platinum, palladium-iron combinations. The catalyst systems containing three active components were composed of either iridium-manganese-iron, iridium-iron-cobalt, or nickel-lanthanum-boron. The frequency of the active metal components was in the order of Pd > Pt > Ni. In addition to palladium and platinum, nickel was also seen, which is used as a common catalyst in industrial practice (Figure 9 right). Regarding the catalyst carrier, we mainly identified metal oxides (zirconium, chromium, titanium, aluminum, and silicon), ferrites, maghemites, zeolites, and activated carbon as typical in the chemical industry. Occasionally, PVP-based catalysts were also investigated [14].

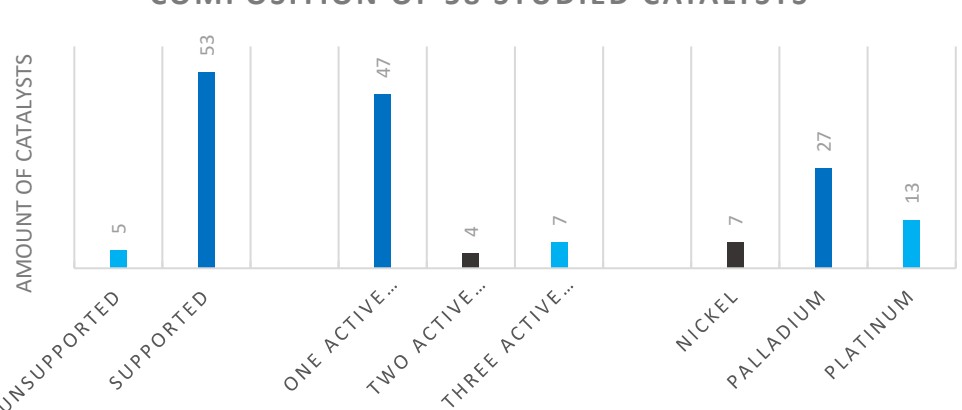

**Figure 9.** Composition of studied catalysts according to support, active component.

### 3.2. Catalyst Ranking and Characterization

Based on the available library of the catalyst data, it is difficult to get a consistent picture of DNT hydrogenation catalysts. However, these catalysts can be well-qualified and comparable, according to the MIRA21 model. A total of 58 catalysts from 15 articles reporting research results were successfully analyzed [14,16,40,44–55].

The catalysts were characterized in detail, as 10 or more known parameters could be collected in each case (from 15). The tested reaction conditions are in the range 295–393 K and 1–50 atm, with the exception of two cases (98 and 150 atm). The time required for maximum conversion ranged from a few minutes to a 1-day interval, which therefore shows a large standard deviation. The average reaction time for 100% conversion is 60 min (Figure 10). This shows that the reaction times of the best catalysts were under 40 min.

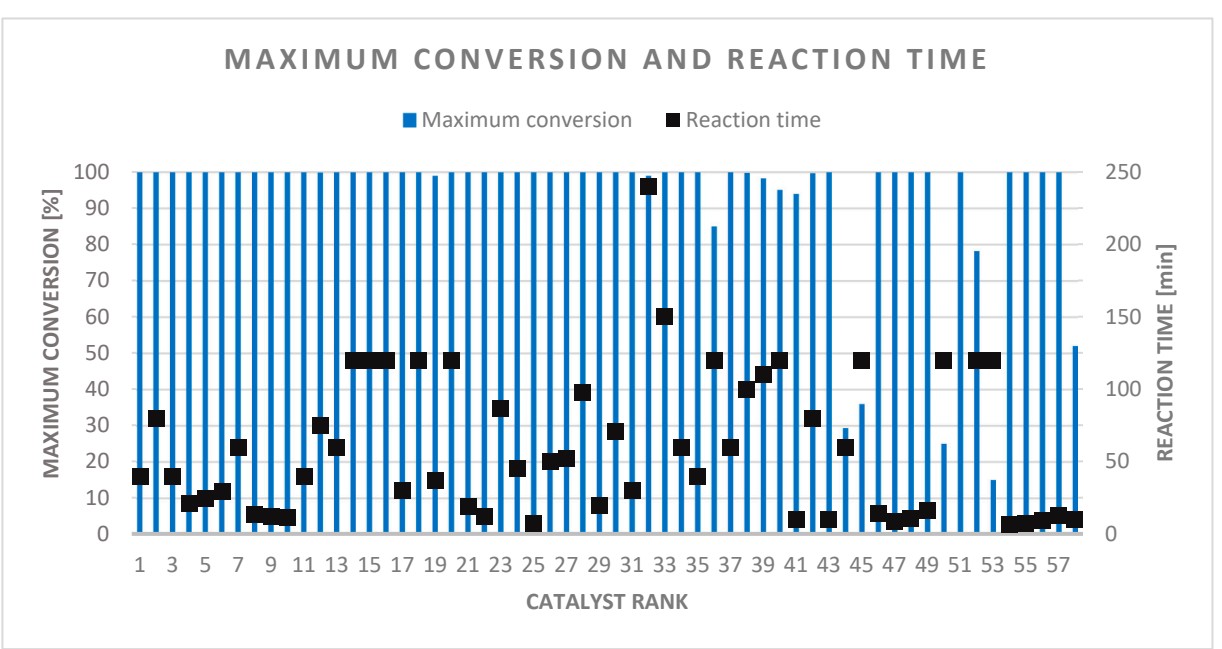

**Figure 10.** Maximum conversion with required reaction time.

The amount of initial dinitrotoluene was in the range of 0.002 and 0.3 mol. The amount of active metal in the catalyst also showed a large deviation from $5.13 \times 10^{-7}$ mol to 0.034 mol. Despite the low amount of the catalyst, as mentioned above, 100% conversion was achieved [54]. The increased amount of the material was typical for nickel-type catalysts.

Furthermore, Figure 11 shows the catalytic performance results for the selected, studied, characterized, ranked, and classified catalysts. The conversion of the studied catalysts in classes D1-Q1-Q2 is over 99 n/n%, however, the product selectivity is much more differentiated. Based on these results, it can be said that achieving the pure TDA product produced during hydrogenation is a serious challenge for researchers. The worst-performing catalysts (class Q4) worked below 50 n/n%.

## CONVERSION AND SELECTIVITY DEPENDS ON CLASSIFICATION

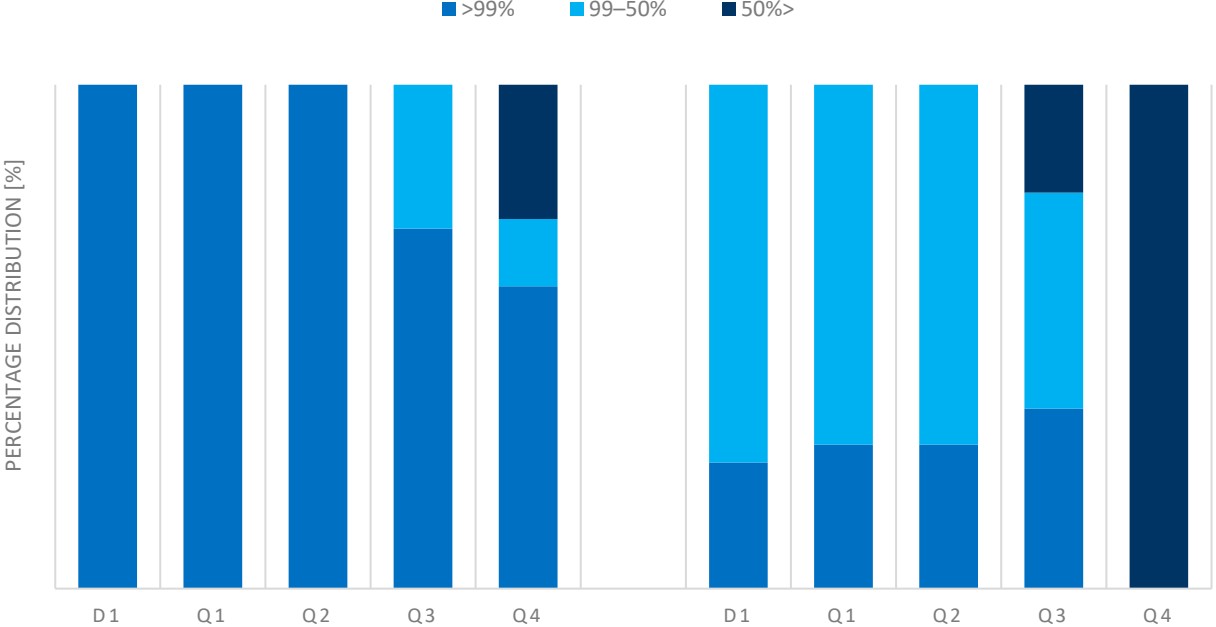

**Figure 11.** Catalytic performance of catalysts.

The catalyst composition changed according to the ranking of the MIRA21 model. In addition, Figure 12 shows the active components and support types of the catalyst systems based on their classification. The best-performing catalysts (class D1) consist of palladium or platinum and transition metal oxide supports. Although nickel is more commonly used in the industry, these types of catalysts are in the lower half of the ranking. Iridium as an active component in the catalyst also obtained a relatively good MIRA21 number. Most of the unsupported, carbon black, $Al_2O_3$ and $SiO_2$-supported catalysts are in the lower half of the ranking.

Practically, the catalyst carrier of the system differed according to the MIRA21 classes. Mainly activated carbon supports can be found in Q2 and Q4 classes. The catalysts with the transition metal supports are at the top of the ranking.

The eight best D1-rated catalysts are listed in Table 1. The columns contain the ID code and the designation of the catalysts, the type of catalyst support and active component, the number of known parameters, and the calculated MIRA21 number. The best MIRA catalysts consist of only one active component and transition metal oxide supports. Based on the results, the platinum-containing catalysts produced better results than their competitors did. The synergistic effect of the combination of the active components is difficult to assess because there is not enough information available. Class D1 includes the catalysts that are studied according to sustainability considerations, such as stability and reactivation capabilities. These catalysts are at the beginning of the innovation pathway and are not yet suitable for industrial application. If the results were compared with the ranking of the catalysts analyzed in the case of the nitrobenzene hydrogenation reaction, it can be found that the best MIRA21-ranked catalyst was similar to the $Pt/ZrO_2$ catalyst, which is one of the most effective catalyst systems in the first class. Zhang et al. prepared a $Pt/ZrO_2/SBA$-

15 hybrid nanostructure catalyst that showed an excellent catalytic performance at 313 K, 7 atm in 50 min for the hydrogenation of nitrobenzene to aniline [56]. They found that the dispersion of $ZrO_2$ in SBA-15 improved the performance of the catalyst due to its mesoporous structure. Therefore, it would be worthwhile to try this catalyst for the synthesis of TDA as well.

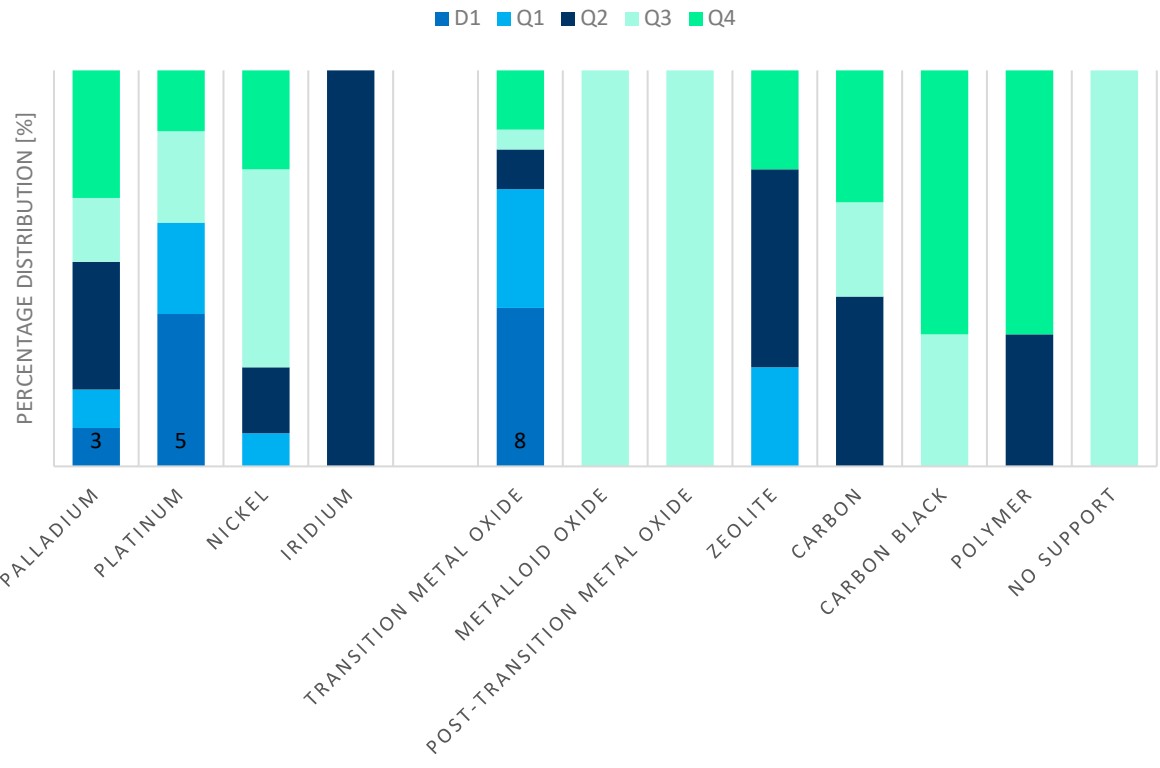

**Figure 12.** Distribution and active components of catalysts according to MIRA21 ranking and classification (D1-best, Q4-worst qualification, according to MIRA21 coloring).

**Table 1.** TOP10 catalyst of MIRA21 ranking.

| | | | D1 CATALYSTS | | | |
|---|---|---|---|---|---|---|
| No. | CATALYST ID | Catalyst Name | Support | Active Component | Number of Known Parameters | MIRA21 Number |
| 1 | HDNT/MIS/2021/2/2 | $Pt/CrO_2$ | Chromium(IV)-dioxide | Platinum | 15 | 11.50 |
| 2 | HDNT/MIS/2021/2/1 | $Pd/CrO_2$ | Chromium(IV)-dioxide | Palladium | 15 | 11.49 |
| 3 | HDNT/MIS/2021/3/1 | $Pd/NiFe_2O_4$ | Nickel ferrite | Palladium | 15 | 11.45 |
| 4 | HDNT/TIA/2020/1/3 | $15Pt/ZrO_2$-300 | Zirconium-dioxide | Platinum | 13 | 11.44 |
| 5 | HDNT/TIA/2020/1/4 | $15Pt/ZrO_2$-400 | Zirconium-dioxide | Platinum | 13 | 11.43 |
| 6 | HDNT/TIA/2020/1/2 | $15Pt/ZrO_2$-200 | Zirconium-dioxide | Platinum | 13 | 11.42 |
| 7 | HDNT/MIS/2021/1/2 | Pd/maghemite | Maghemite | Palladium | 15 | 11.35 |
| 8 | HDNT/TIA/2020/1/5 | $45Pt/ZrO_2$-300 | Zirconium-dioxide | Platinum | 13 | 11.06 |

The work of Hajdu et al. focused on the development of new magnetic catalysts for the hydrogenation of DNT to TDA [44,53,55]. One of the catalysts is Pd/NiFe2O4, which has achieved 99 n/n% TDA yield at 333 K and 20 atm. In this work, they synthetized the nickel ferrite spinel nanoparticles to solve the problem of separating the catalyst from the products by magnetization. Another magnetic catalyst with good catalytic performance is Pd/maghemite, which is made by a combustion method with a sonochemical step. Palladium on a maghemite support resulted in a high catalytic activity for TDA synthesis at about 60 min and under the same reaction conditions as ferrite hydrogenation. The first

and the fifth place of the MIRA rankings were the chromium oxide platinum and palladium catalysts. These innovative systems yielded excellent performing catalysts. It was prepared with chromium (IV) oxide nanowires that were decorated with platinum and palladium nanoparticles. These catalysts showed high catalytic activity at 333 K and 20 atm. If a Pt/CrO$_2$ catalyst was used, 304.8 mol of TDA was produced under these conditions, while only 1 mol of the precious metal catalyst was used. When palladium is used as an active component, only 60.14 mol TDA was produced, but it is also a relatively large amount. From an industrial point of view, it is important that this type of catalyst could be easily separated from the reaction mixture due to its magnetic properties. The stability of the catalyst was studied, and it was found that the catalyst could be used at least four times without regeneration.

Ren and his colleagues made half of the D1 class catalysts, and these catalytic systems consisted of zirconium oxide supports and platinum precious metal [54]. Ren et al. prepared the ZrO$_2$-supported platinum catalysts with different Pt concentrations and at different reduction temperatures. They found that the 0.156% Pt-containing zirconium oxide catalyst has the highest catalytic performance at 353 K and 20 atm. According to their results, the use of this catalyst reached an initial hydrogen consumption of 4583 mol H2 mol Pt-1 min-1. In this work, they investigated the interaction between the precious metal and the oxide support. It was found that zirconium oxide had the highest adsorption capacity for platinum ions due to its ability to be protonated and deprotonated.

## 4. MIRA21 Method

In our previous work, we successfully developed the Miskolc Ranking 2021 (MIRA21) system as a multistep process for the identification of new and useful patterns in the catalyst data sets to provide a standard algorithm for catalyst characterization and to compare and rank catalysts with minimal bias [24]. It is a practical and functional mathematical model of exact catalyst qualification with four classes of descriptors: catalyst performance, reaction conditions, catalyst conditions, and catalyst sustainability. The comparison of TDA catalysts could enable the supporting design of catalysts and the monitoring of research and development trends. The model facilitates the determination of the direction of catalyst development by establishing a system for ranking and classifying the catalysts. Furthermore, the standardization of the data in scientific publications could also benefit from accurate and coherent data. Figure 13 illustrates the process of the MIRA21 method from the literature sources to useful knowledge.

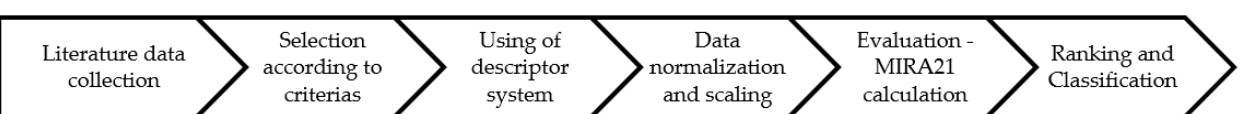

**Figure 13.** Process steps of MIRA21 model.

Table 2 shows the descriptor system of the model. The parameters can be divided into four classes with different weighting coefficients. The catalyst performance class includes conversion, selectivity, yield, and turnover number attributes. The second group contains the reaction conditions of the laboratory or large-scale experiments with the prepared catalysts. The third group is the catalyst conditions, with these two easily described parameters included in it. The last group addresses the sustainability and industrial application of the catalysts.

**Table 2.** Descriptor system of MIRA21 for TDA synthesis [24].

| | Categories | | No. | Notation | Name | Unit | Definition |
|---|---|---|---|---|---|---|---|
| **Quantifiable parameters** | **Catalyst performance** | **I.** | 1. | $X_{PRmax}$ | Maximum conversion | n/n% | Maximum desired product conversion achieved on a given catalyst |
| | | | 2. | $Y_{PR}$ | Product yield | n/n% | Product yield for maximum conversion |
| | | | 3. | $S_{PR}$ | Product selectivity | n/n% | Product selectivity for maximum conversion |
| | | | 4. | $TON_{PR}$ | Turnover number | - | Number of moles of product formed per 1 mol active metal when the maximum conversion reached |
| | **Reaction conditions** | **II.** | 5. | $Tmax_{conv.}$ | Temperature | K | Reaction temperature for maximum conversion |
| | | | 6. | $P_{max.conv.}$ | Pressure | atm | Reaction pressure for maximum conversion |
| | | | 7. | $t_{max.conv.}$ | Time | min | Time required to reach maximum conversion |
| | | | 8. | $n_{cat.}$ | Molar amount of initial catalyst | mol | The molar amount of the active metal involved in the reaction; in case of several metals, the sum of molar numbers |
| | | | 9. | $n_{start}$ | Molar amount of starting reagent | mol | The initial amount of starting reagent involved in the reaction |
| | **Catalyst conditions** | **III.** | 10. | CPZ | Catalyst particle size | nm | Average particle size of the catalyst |
| | | | 11. | CSA | Catalyst surface area | $m^2/g$ | Catalyst (active metal + support) surface area |
| **Does the publication contain information about these subjects?** | | | | | | | |
| **Non-quantifiable parameters** | **Sustainable parameters** | **IV.** | 12. | Rea | Information about reactivation | - | Reactivation means the physical process by which the activity of the catalyst used returns to or near the original activity level. |
| | | | 13. | Stab | Information about stability of catalyst | - | Stability means preservation of catalytic activity |
| | | | 14. | Care | Information about catalyst carrier effect | - | Carrier effect means the catalyst support influences the catalytic reaction |
| | | | 15. | | Catalyst carrier effect | - | Nature of the effect (positive, no effect, negative) |

Table 2 describes the parameters used to characterize the performance of the catalyst, as well as the main mathematical equations used in the calculation of the MIRA21 number and ranking.

Figure 14 lists the equations used in the MIRA21 model, where $n_{DNT}$, $n_{TDA}$, and $n_{catalyst}$ are the corresponding molar amounts of the compounds; A is the value of the attribute; $A^t$ is the transformed attribute value; $min_A$ and $max_A$ are the corresponding calculated minimum and maximum values of the attribute in the data set; MIN is the minimum scoring point; MAX is the maximum scoring point; i = 1 ... 15 is the number of attributes; MAXrank is the highest and MINrank is the lowest score of the MIRA21 ranking.

| Used Equations | |
|---|---|
| Maximum Conversion | $X_{Prmax}\% = \dfrac{consumed\ n_{DNT}}{initial\ n_{DNT}} * 100$ |
| Product Yield | $Y_{PR}\% = \dfrac{synthetized\ n_{TDA}}{initial\ n_{DNT}} * 100$ |
| Product Selectivity | $S_{PR}\% = \dfrac{synthetized\ n_{TDA}}{consumed\ n_{DNT}} * 100$ |
| TurnOver Number | $TON_{PR} = \dfrac{synthetized\ n_{TDA}}{n_{catalyst}}$ |
| Normalization | $A^t = MIN + \dfrac{(MAX - MIN) * A - min_A}{max_A - min_A}$ |
| MIRA21 | $MIRA21 = log \displaystyle\prod_{i=1}^{n} A_i^t \qquad i = 1\ ....15$ |
| Classification | $SLofD1class = MAXrank - \left(\dfrac{MAXrank - MINrank}{10}\right)$ |

**Figure 14.** MIRA21 used equations [24].

## 5. Summary and Conclusions

In summary, Table 3 includes the MIRA21 results and the classification of selected and studied catalysts for the hydrogenation of DNT. The aim of this work is to make an overview of the hydrogenation of dinitrotoluene to toluenediamine. The chemical technology, development of reaction mechanism, and previous catalyst research were summarized by a quantitative comparison method called MIRA21. In total, 58 catalysts from 15 research articles were selected and studied with the MIRA21 model, which covered the complete scientific literature of the catalytic hydrogenation DNT. According to the ranking and classification, eight catalysts were ranked in the highest class (D1).

**Table 3.** Catalysts for dinitrotoluene hydrogenation to toluenediamine in liquid phase [14,16,40,44–55].

| CATALYSTS | | | | | | | |
|---|---|---|---|---|---|---|---|
| No. | CATALYST ID | Catalyst Name | Catalyst Support | Main Active Component | Known Par. | MIRA21 Number | Class |
| 1 | HDNT/MIS/2021/2/2 | $Pt/CrO_2$ | Chromium(IV)-dioxide | Platinum | 15 | 11.50 | D1 |
| 2 | HDNT/MIS/2021/2/1 | $Pd/CrO_2$ | Chromium(IV)-dioxide | Palladium | 15 | 11.49 | D1 |
| 3 | HDNT/MIS/2021/3/1 | $Pd/NiFe_2O_4$ | Nickel ferrite | Palladium | 15 | 11.45 | D1 |
| 4 | HDNT/TIA/2020/1/3 | $15Pt/ZrO_2$-300 | Zirconium-dioxide | Platinum | 13 | 11.44 | D1 |
| 5 | HDNT/TIA/2020/1/4 | $15Pt/ZrO_2$-400 | Zirconium-dioxide | Platinum | 13 | 11.43 | D1 |
| 6 | HDNT/TIA/2020/1/2 | $15Pt/ZrO_2$-200 | Zirconium-dioxide | Platinum | 13 | 11.42 | D1 |
| 7 | HDNT/MIS/2021/1/2 | Pd/maghemite | Maghemite | Palladium | 15 | 11.35 | D1 |
| 8 | HDNT/TIA/2020/1/5 | $45Pt/ZrO_2$-300 | Zirconium-dioxide | Platinum | 13 | 11.06 | D1 |

**Table 3.** *Cont.*

| | | | | | | | |
|---|---|---|---|---|---|---|---|
| **CATALYSTS** | | | | | | | |
| No. | CATALYST ID | Catalyst Name | Catalyst Support | Main Active Component | Known Par. | MIRA21 Number | Class |
| 9 | HDNT/TIA/2020/1/6 | 60Pt/ZrO$_2$-300 | Zirconium-dioxide | Platinum | 13 | 11.01 | Q1 |
| 10 | HDNT/TIA/2020/1/7 | 85Pt/ZrO$_2$-300 | Zirconium-dioxide | Platinum | 13 | 11.00 | Q1 |
| 11 | HDNT/MIS/2021/3/2 | Pd/CoFe$_2$O$_4$ | Cobalt ferrite | Palladium | 15 | 10.84 | Q1 |
| 12 | HDNT/SHA/2012/1/1 | Ni/HY catalyst | HY molecular sieve | Nickel | 15 | 10.77 | Q1 |
| 13 | HDNT/MIS/2021/1/1 | Pt/maghemite | Maghemite | Platinum | 15 | 10.67 | Q1 |
| 14 | HDNT/MIS/2021/3/3 | Pd/CuFe$_2$O$_4$ | Copper ferrite | Palladium | 15 | 10.48 | Q1 |
| 15 | HDNT/MIS/2022/1/3 | Pd/NiFe$_2$O$_4$-NH2 | Nickel-ferrite | Palladium | 13 | 10.31 | Q1 |
| 16 | HDNT/MIS/2022/1/1 | Pd/CoFe$_2$O$_4$-NH2 | Cobalt-ferrite | Palladium | 13 | 10.28 | Q2 |
| 17 | HDNT/MIS/2021/1/3 | Pd-Pt/maghemite | Maghemite | Palladium | 14 | 10.14 | Q2 |
| 18 | HDNT/PUN/1999/1/5 | 20% Ni/HY | HY zeolite | Nickel | 14 | 9.50 | Q2 |
| 19 | HDNT/DAL/1997/1/2 | PVP-Pd-1/4 Pt | PVP | Palladium | 13 | 9.47 | Q2 |
| 20 | HDNT/PUN/1999/1/6 | 10% Ni/HY | HY zeolite | Nickel | 14 | 9.44 | Q2 |
| 21 | HDNT/MES/2001/1/1 | MGPd05 | Chemviron SC XII active carbon | Palladium | 13 | 9.24 | Q2 |
| 22 | HDNT/MES/2001/1/3 | MGPd1b | Chemviron SC XII active carbon | Palladium | 13 | 9.23 | Q2 |
| 23 | HDNT/MES/2001/1/8 | MGPd5a | Chemviron SC XII active carbon | Palladium | 13 | 9.20 | Q2 |
| 24 | HDNT/HAN/2001/1/2 | B | Chemically activated carbon | Iridium | 13 | 9.19 | Q2 |
| 25 | HDNT/MES/2001/1/4 | MGPd1c | Chemviron SC XII active carbon | Palladium | 13 | 9.19 | Q2 |
| 26 | HDNT/HAN/2001/1/1 | A | Chemically activated carbon | Iridium | 13 | 9.19 | Q2 |
| 27 | HDNT/MES/2001/1/7 | MGPd5 | Chemviron SC XII active carbon | Palladium | 13 | 9.17 | Q2 |
| 28 | HDNT/MES/2001/1/5 | MGPd1d | Chemviron SC XII active carbon | Palladium | 13 | 9.14 | Q2 |
| 29 | HDNT/MES/2001/1/2 | MGPd1a | Chemviron SC XII active carbon | Palladium | 13 | 9.13 | Q2 |
| 30 | HDNT/MES/2001/1/6 | MGPd3 | Chemviron SC XII active carbon | Palladium | 13 | 9.06 | Q3 |
| 31 | HDNT/HAN/2001/1/3 | C | Steam activated carbon | Palladium | 13 | 9.03 | Q3 |
| 32 | HDNT/MIS/2022/1/2 | Pd/CdFe$_2$O$_4$-NH$_2$ | Cadmium-ferrite | Palladium | 13 | 8.95 | Q3 |
| 33 | HDNT/ZUR/1987/1/1 | 0.5 % Pt/Al$_2$O$_3$ | Al$_2$O$_3$ | Platinum | 13 | 8.88 | Q3 |
| 34 | HDNT/HAN/2001/1/4 | D | Steam activated carbon | Palladium | 13 | 8.79 | Q3 |
| 35 | HDNT/HAN/2001/1/5 | E | Oleophilic carbon black | Palladium | 13 | 8.78 | Q3 |
| 36 | HDNT/PUN/1999/1/2 | 20% Ni/SiO$_2$ | SiO$_2$ | Nickel | 14 | 8.63 | Q3 |
| 37 | HDNT/SHA/2012/1/4 | Ni-La6-B | | Nickel | 12 | 8.36 | Q3 |
| 38 | HDNT/SHA/2012/1/3 | Ni-La4-B | | Nickel | 12 | 8.34 | Q3 |
| 39 | HDNT/SHA/2012/1/2 | Ni-La2-B | | Nickel | 12 | 8.33 | Q3 |
| 40 | HDNT/SHA/2012/1/1 | Ni-La0-B | | Nickel | 12 | 8.32 | Q3 |
| 41 | HDNT/SAP/2004/1/3 | Pt/C in ethanol | Active carbon | Platinum | 13 | 8.02 | Q3 |
| 42 | HDNT/SHA/2012/1/5 | Ni-La8-B | | Nickel | 12 | 7.92 | Q3 |
| 43 | HDNT/SAP/2004/1/2 | Pt/C in ethanol | Active carbon | Platinum | 13 | 7.90 | Q3 |
| 44 | HDNT/TIA/2020/1/1 | 15Pt/ZrO$_2$-100 | Zirconium-dioxide | Platinum | 13 | 7.85 | Q4 |
| 45 | HDNT/PUN/1999/1/4 | 20% Ni/HZSM-5 | HZSM-5 | Nickel | 14 | 7.75 | Q4 |
| 46 | HDNT/TAE/1993/1/1 | SA | Activated carbon | Palladium | 11 | 7.58 | Q4 |
| 47 | HDNT/TAE/1993/1/3 | DA | Activated carbon | Palladium | 11 | 7.52 | Q4 |
| 48 | HDNT/TAE/1993/1/2 | SAON | Activated carbon | Palladium | 11 | 7.50 | Q4 |
| 49 | HDNT/TAE/1993/1/4 | DAON | Activated carbon | Palladium | 11 | 7.49 | Q4 |
| 50 | HDNT/PUN/1999/1/1 | 20% Ni/Al$_2$O$_3$ | Al$_2$O$_3$ | Nickel | 14 | 7.43 | Q4 |
| 51 | HDNT/BAR/2000/1/1 | Pd(AAEMA)2/EMA/EGDMA | Polymer-supported complex | Palladium | 13 | 7.27 | Q4 |
| 52 | HDNT/DAL/1997/1/1 | PVP-PdCl2 | PVP | Palladium | 10 | 6.99 | Q4 |
| 53 | HDNT/PUN/1999/1/3 | 20% Ni/TiO$_2$ | TiO$_2$ | Nickel | 14 | 6.95 | Q4 |
| 54 | HDNT/TAE/1993/1/7 | VB | Carbon black | Palladium | 10 | 6.80 | Q4 |
| 55 | HDNT/TAE/1993/1/8 | VON | Carbon black | Palladium | 10 | 6.80 | Q4 |
| 56 | HDNT/TAE/1993/1/6 | DAOH | Activated carbon | Palladium | 10 | 6.80 | Q4 |
| 57 | HDNT/TAE/1993/1/5 | DAOS | Activated carbon | Palladium | 10 | 6.80 | Q4 |
| 58 | HDNT/SAP/2004/1/1 | Pt/C in scCO$_2$ | Active carbon | Platinum | 13 | 6.68 | Q4 |

The number of catalysts developed for TDA synthesis is low, since the scientific research focused mostly on the reaction mechanism and reaction kinetic. Despite this fact, many different catalyst systems have been developed.

More than 80% of the 58 types of catalysts produced and tested had excellent conversions, but only 45% of them demonstrated a selectivity above 90% n/n%. More than 80% of the produced catalysts consisted of only one active component. Since the combination of catalysts has not been scarcely investigated, one recommended direction of research is the multi-component catalyst. Catalyst development represents a new trend that has led to

the establishment of many high-performance catalysts. Based on the analyzed catalysts, compared to the traditional carbon-based supports, catalysts with oxide and/or magnetic supports showed better results in laboratory conditions. Carbon-supported nickel catalysts are primarily used in the industry, but nickel catalysts did not yield the best results. The advantage of well-performing magnetic catalysts due to their ability to be repaired is indisputable, but the economic implications of their industrial application must also be considered.

**Author Contributions:** Conceptualization, A.J.-N. and B.V.; methodology, A.J.-N., B.V., V.H. and L.V.; writing—original draft preparation, A.J.-N., V.H., L.F. and L.V.; writing—review and editing, A.J.-N., L.F. and B.V.; supervision, B.V.; funding acquisition, B.V. All authors have read and agreed to the published version of the manuscript.

**Funding:** This research was supported by the Ministry of Innovation and Technology-financed 2020-1.1.2-PICI-KFI-2020-00121 project. This research was prepared with the professional support of the Doctoral Student Scholarship Program of the Cooperative Doctoral Program of the Ministry of Innovation and Technology financed from the National Research, Development and Innovation Fund.

**Acknowledgments:** We thank Tamás Purzsa from Wanhua-BorsodChem for his helpful contributions. We also acknowledge the opportunity provided by Wanhua-BorsodChem to conduct this study.

**Conflicts of Interest:** The authors declare no conflict of interest.

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
