# Peer review of "Overview of Catalysts with MIRA21 Model in Heterogeneous Catalytic Hydrogenation of 2,4-Dinitrotoluene"

_catalysts, doi:10.3390/catal13020387_

Round 1

Reviewer 1 Report

Comments

Suggestion for Manuscript ID: catalysts-2213723

The present work reviewed Catalysts with MIRA21 Model in Heterogeneous 2
Catalytic Hydrogenation of 2,4-Dinitrotoluene. aim of the work is to characterize, rank, and compare the catalysts of 2,4-dinitrotoluene catalytic hydrogenation to 2,4-toluenediamine by applying the MIskolc RAnking 21 (MIRA21) model. This systematic overview provides a comprehensive picture of the reaction, technological process, and the previous and newest research results. Over all, this manuscript was organized well and the research significance of this work was highlighted. However, there are still some questions that need to be appropriately revised before it can be published. My suggestions are as the follows:

1. Please keep the format uniform. Such as line 35, Toluene diisocyanate should be abbreviated with TDI. Line 72, toluene-diisocyanate, line 75, toluene diisocyanate, et al.

2. Section2, about the technological process, the effect of temperature and pressure on yield should be reviewed and discussed.

3. Section 3, The effect of catalyst on reaction rate should also be summarized, and the data about the reaction rate and yield should be summarized in a table.

4. line 157, dimethyl-2-nitrobenzene. compound G, where comma (,) should be in the middle?

5. The number of references is only 51, which is very small for review articles.

6. The references should be unified, such as Ref. [18], [28], [44].. etc. without pages.

Reviewer 2 Report

The work "Overview of Catalysts with MIRA21 Model in Heterogeneous Catalytic Hydrogenation of 2,4-Dinitrotoluene" reviews the synthesis of 2,4-DNT employing MIRA21. Overall, the work is interesting, very well written and organised. Results are correctly described. I suggest its acceptance after  addressing the following issues:

- I detected some errors: Table 1, maybe because of the format, category III "catalyst...?" there is a missing word here.

- lack of subscripts on molecular formulas (ej Table 3).

- The introduction might be also improved by including other recent studies involving amino/nitro-aromatic compounds (see for instance Sciscenko et al 2021, A Rational Analysis on Key Parameters Ruling Zerovalent Iron-Based Treatment Trains: Towards the Separation of Reductive from Oxidative Phases, Nanomaterials (11) 2948).

- Please, give further details of MIRA21 for non-experts or give further references in the introduction. Moreover, why was not "cost" or "generated residues" employed in the ranking parameters? This is certainly of interest when looking at which process to choose. 
